# Temples and bats in a homogeneous agriculture landscape: Importance of microhabitat availability, disturbance and land use for bat conservation

**T. Ganesh** ⓘ *, **A. Saravanan, M. Mathivanan** ⓘ

Ashoka Trust for Research in Ecology and the Environment (ATREE), Royal Enclave, Bangalore, Karnataka state, India

* tganesh@atree.org

**Data Availability Statement:** The data used in the paper is available at https://doi.org/10.6084/m9.figshare.20059091.v1.

## Abstract

Cave-dwelling bats widely use anthropogenic structures such as temples in south Asia as roosting and nursery sites. Such roosts are constantly under threat, even more so after the COVID-19 pandemic. Despite the importance of such roosts, there is no detailed understanding of what makes temples favorable for bats and the critical factors for their persistence. Here we relate temple microhabitat characteristics and land use around ancient temples (>400 years) to bat species richness and abundance in the Tamiraparani river basin of south India. Temples were selected for sampling along the river basin based on logistics and permission to access them. We counted bats at the roost in the mornings and late afternoons from inside the temples. Temple characteristics such as dark rooms, walkways, crevices, towers, and disturbances to the roosts were recorded. Based on European Space Agency land use classifications, we recorded land use such as crops, trees, scrub, grassland, urban areas, and water availability within a 5 km radius of the temple. Generalized Linear Mixed Models were used to relate the counts in temples with microhabitats and land use. We sampled 59 temples repeatedly across 5 years which yielded a sample of 246 survey events. The total number of bats counted was 20,211, of which *Hipposideros speoris* was the most common (9,715), followed by *Rousettus leschenaultii* (5,306), *Taphozous melanopogon* (3,196), *Megaderma lyra* (1,497), *Tadarida aegyptiaca* (303), *Pipistrellus sp.* (144) and *Rhinopoma hardwickii* (50). About 39% of the total bats occurred in dark rooms and 51% along walkways. Species richness and total abundance were related to the availability of dark rooms and the number of buildings in the temple. Land use elements only had a weak effect, but scrub and grassland, even though they were few, are critical for bats. We conclude that retaining undisturbed dark rooms with small exits in temples and other dimly lit areas and having natural areas around temples are vital for bat conservation.

**Funding:** The work was partly supported by a grant from Bat Conservation International Global Grassroots Conservation Fund. We thank Maria, Lise and K.S.Seshadri for help in the field. We are grateful to Dr.R.Ganesan, Dr.M.Soubadra Devy for their help during the initial stages of the work. We also tank Rajkamal, Seshadri, Prashanth and Soubadra for their comments on the MS. We thank Thamizhazhagan for helping in capturing bats and Muthupandi for assistance in the field. Abhishek helped us with landuse analysis and Thalavaipandi drew the map. The Temple authorities and temple staffs in both Tirunelveli and Thoothukudi districts permitted us to do the study. Jermiah Rajanesan, director Dhonavur Fellowship and Maria Joseph, Head Master, Christuraja Hr., Sec., School, Palayamkottai helped us with logistics and we are grateful to them for their help. There was no additional external funding received for this study.

**Competing interests:** The authors have declared that no competing interests exist.

## Introduction

Many studies suggest that human-modified landscapes with a heterogeneous mosaic of different land uses can preserve species-rich bat assemblages [1–7]. This is possible with the concurrent availability of microhabitats that bats use as roosting sites ranging from caves, crevices, tree cavities, foliage, and several types of anthropogenic structures, including road bridges and buildings [8, 9]. Although these microhabitats are important features of a bat's environment, the selection of specific roost sites can strongly impact the survival and fitness of bats [10–12]. However, emerging challenges such as changing agricultural practices and climate change pose significant threats to bat assemblages and populations in human-dominated landscapes [13, 14]. They could negatively impact human populations by altering the ecosystem services provided by bats [15, 16]. Moreover, bats that roost in anthropogenic structures are vulnerable to disturbance since, just like cave roosting species, they often form large, concentrated aggregations attracting attention [17]. Most studies on bats using anthropogenic structures have come from urban areas and under temperate conditions [9, 15, 18–20]. While bats are still poorly studied in the Paleotropics [21], the selection of microhabitats and landscape-level influences on their population remains unknown [22–24]. Moreover, most studies on bat–habitat relationships are conducted at fixed spatial scales [25], while recent studies have shown the importance of scale for bats [26, 27].

In South Asia, several species of both insectivorous and frugivorous bats use temples as roosting sites [28, 29]. In south India, numerous very old (>400y) temples dot the landscape and form important bat habitats [30]. These temples are built of granite stones with several enclosures and have towers (*Gopuram*) made of bricks. The temples are not air-conditioned, and the temperature inside is stable and warm (320–330 C) and relatively constant across the day (personal obs T.Ganesh). Bats use these temples as roost and nursery and enter and leave through numerous open vents that circulate air inside the temple.

Bat use of anthropogenic structures, such as temples, brings them into conflict with humans and makes bat conservation in the human-dominated landscape a challenge. These temples dominate the human-agricultural landscape where water-intensive paddy and banana cultivation have replaced traditional crops. Such agricultural practices are considered a major threat to bats [14]. With irrigated agriculture spreading to dry regions and with a predicted increase in rains [31], threats to the bat population can rise in the future. Bats in many temples have dwindled [32, 33], and many are driven away during temple renovation [34]. The changing attitude of people towards bats has led many temples to be made "bat-proof" by placing nets and power washing roost areas with chlorinated chemicals to prevent bats from colonizing. Further, the recent COVID-19 pandemic has made them more vulnerable to disturbance and direct persecution. In India, only two out of 120 species of bats come under the strict Wildlife Protection Act 1972, while others under the International Union for Conservation and Nature (IUCN) category remain unprotected.

Management decisions for bats and buildings, including the timing of maintenance activities, restriction of human use, and bat deterrence or exclusion, require information on the roosting habits of temple-roosting bats. In addition, studies in temperate regions have shown how features of bridges and buildings can help identify sites for conservation and thereby reduce bat-human conflict [9, 20]. Such efforts to conserve bats in tropical regions are needed. It's therefore essential to understand what attributes of temples are critical for bats and how such information can help bat conservation efforts in temples.

More than 50% of the 35 bat species found in Tamil Nadu state dwell in caves and old buildings [28]. All bats are insectivores, except *Rousettus leschenaultii*, a frugivore. In the paddy agriculture-dominated landscape of south India, ancient temples are often the only habitat for

several bat species. They provide critical ecosystem services for agriculture and, in south Asia, can save up to US$1.2 million per year from insect depredation [35] and consume large quantities of mosquitoes, thus providing human health benefits [36]. However, despite the importance of bats in providing key ecosystem services in human-dominated landscapes [34, 37], no systematic study on the ecology of bats in temples exists, while behavioral studies have been carried out on them in South India [38, 39]. Moreover, with populations of bats dwindling, no attempt to monitor bats in the temples or how important temples are in an agricultural matrix to bats has been investigated. Much of the bat diversity in human-dominated landscapes can be attributed to habitat diversity and roost site availability in an area. However, in tropical agricultural regions with rice paddies and bananas dominating the landscape providing low habitat heterogeneity, it would be interesting to compare the role of land use elements and availability of microhabitats on bat species richness and abundances.

We initiated a bat monitoring program in 2012 in temples of south India to understand why bats use temples and how the population is responding to disturbance and land use. In this study, we identify temples used by bats for roosting in a semi-arid region characterized by the winter monsoon (Oct to Dec) and agriculture dominated by irrigated rice paddy and banana for 6–8 months a year. The rest of the time, the land is fallow in most places. We recorded the species composition, rarity, and abundance of bats in temples and identified factors that affect bat colonization, such as microhabitat availability, disturbance in temples and landscape features surrounding the temples. This paper tests three major hypotheses: 1. Disturbance at the roost negatively affects species richness and abundance of bats, 2. Roost characteristics are more important than landscape features for bats to colonize temples, and 3. In a homogeneous monoculture cropping, species occurrence and abundance are independent of the spatial availability of habitats around roosts.

## Materials and methods

### Study area

The study was conducted between lat 8.739, long 77.443 and lat 8.306, long 77.913 in Tirunelveli & Thoothukudi districts of Tamil Nadu state in South India. The semi-arid region experiences the northeast winter monsoons between Oct-Dec while the rest of the months are dry. Several ancient temples built during the Chola and Pandya dynasties 500 to 1000 years back exist in the districts along the perennial Tamiraparani river and its tributaries. The river and numerous interconnected village ponds are the lifelines for paddy and banana-dominated agriculture, supporting a dense human population in the region. The ancient temples are built of large granite stones and provide a habitat for several bat species. There are over 4041 temples [40] in the study area, with 2–3 villages having one ancient temple. A typical temple structure includes towers at the four cardinal entrances and one over the deity. The deity chamber is lit by an oil lamp and an electric bulb, a walkway surrounding the deity chamber for circumambulation by devotees, and several rooms, including storerooms and a kitchen, which are dark and rarely frequented by people.

**Temple selection.** We selected 59 temples from the Tamiraparani basin and surveyed them with 2–4 personnel between 06:00–12:00 h and again between 16:00 to 17:00 h to detect the presence of bats. The survey was done from May to Dec in 2012, 2013, 2014, 2018 and 2019, totaling 246 survey efforts, with most temples revisited each year. All species of *Yinpterochiroptera* and *Yangochiroptera* bats found roosting inside the temples were recorded. First, the presence of bats in the temple was confirmed by several signs such as droppings and smell, interviewing local people, temple priests, and caretakers. Later, using a red LED light, we scanned the dark corners, temple roof, abandoned & isolated rooms inside the temple and the

temple tower for chiropteran presence. Once bat colonies were detected, we counted the number of individuals, and in some cases, especially in closed buildings, we took photographs to estimate them.

**Temple features.** *a. Microhabitat.* There were multiple microhabitats inside the temples where bats could roost. Dark rooms: closed rooms with few small openings, low natural light and not frequently used by people; Crevices: stone pillars that supported the roof created gaps between the joints for some species to roost; Temple towers: conical structures built over the entrance or the deity and Walkways: wide walking space around the deity room covered by a stone roof. We also counted the number of independent buildings in the temple complex, which bats use to move between them during the day due to disturbance. In 2012 we did a one-time measurement of temperature and humidity while sampling for bats and found no difference between temples and therefore did not consider them in the analysis. We could not set dataloggers for continuous monitoring because of objections by the temple authorities to placing any devices inside temples. We estimated temple size based on visual observation; those with deity rooms and walkways were considered small, those with deity rooms, walkways and towers considered medium-sized, and those having deity rooms, multiple towers, and two walkways (inside and outside) were considered large. We also recorded the number of trees in the temple complex and open wells used by bats.

*b. Disturbance.* We classified disturbances such as renovation, construction, power washing, and other maintenance activities as 0 or 1. A temple received a value of 0 when there was no disturbance and a value of 1 when there was a disturbance. We also collected information on the number of temple visitors and classified temples on a scale of 1 to 3, with 1 indicating few visitors and 3 indicating having large crowds.

## Land-use features

We used the composite analysis of land cover available using high-resolution (10m) European Space Agency (ESRI) data for 2020 [41] to identify major land uses around temples in the region. Unfortunately, cloud cover prevented us from deciphering land use change across each sampling period. Water, scrub, grassland, bare ground, flooded area, crop, urban and tree-covered areas were the primary land use identified [41]. We combined bare ground and grassland into grassland habitat and water and flooded area into water habitat after verifying the land use on the ground and using Google Earth. Crop availability each season was obtained from Google Earth images when available and further substantiated by informal interviews with farmers around each temple. If crops were present, we used the ESRI calculated value of crop area in the analysis, and if not, we considered it zero. The extent of other land uses has not changed much during the sampling period based on the data available with the district administration (Statistical handbook of Tirunelveli district 2009–2019).

Since the spatial scale is a driver for bat foraging [27] we categorized land use at 500m, 1km, 3km and 5km from the temple based on the foraging distance of bats from their roost [38]. We also recorded trees and water availability (open wells) used by bats in the temple complex for land-use analysis at 0km level.

## Species identification and abundance

We identified bats to the species level using the bat identification guide by Bates and Harrison (1997). Bats were counted at the roost when they were least active (late morning and early evenings) using a flashlight with red filters. We took photographs as soon as bat aggregations were noticed in a particular microhabitat and later counted the bats in the picture to estimate abundance [42]. Several images taken in quick succession gave a complete count. We could

not do a roost exit count in the evening as it was not logistically possible to be in each temple at the right time for the count. Binoculars helped to see the species' features for identification.

## Data analysis

We used total roost count (abundance) and total species richness at each temple for the analysis. We constructed a linear model to explain the species richness and abundance of bats in the temples based on the microhabitat and landscape features. We based the variables used in our linear models on our observations made in 2012. We eliminated strongly correlated variables (>0.70) with other variables to reduce variable redundancy and retained ones that were biologically meaningful. We used mixed models to relate the total species richness and abundance of bats with temple variables and landscape elements. Since temples were sampled more than once and repeated measurements on the same temple are often correlated, we accounted for this by using random effects in the mixed models [43]. We used lmer package in R for normally distributed species richness and glmmadmb package with a negative binomial function for overdispersed bat abundances [44]. We considered temple I.D. and year as a random factors in the model and temple characteristics, disturbance and land use elements as fixed factors. We used the function *dredge* in MuMIn package (version 1.43.17) in R to obtain all possible combinations of models using the select variables. We fitted two models, one for temple characteristics with disturbance and the other for land use elements at 5 spatial scales; 0m, 500m, 1km, 3km and 5km. The models were ranked using Akaike's Information Criterion (AICc) corrected for sample size and model weights. Model parameters were averaged for closely related models [45] using model selection criteria in MuMIn package (version 1.43.17) in R. We calculated model-averaged parameter estimates and unconditional standard errors to assess the relative importance of each variable and account for model uncertainty [46]. We only considered parameters with 95% confidence intervals not overlapping with 0 to be informative [47]. We tested for differences in microhabitats and land use between species using Kruskal Wallis and Mann Whitney tests. Data analysis and statistical tests were done using Microsoft Excel®, PAST® and R version 4.

## Results

### Species richness and abundance

The total roosts sampled across 5 years was 351 in 246 temples. Of the 351 temples, 286 had bats and 16% of these had multiple roosts. Seven species of bats were recorded, of which six were Yinpterochiroptera and one *Rousettus leschenaultii* was a Yangochiroptera (Table 1). Among the Yinpterochiroptera, *Hipposideros speoris* was the most frequently seen species found in 135 (38%) roosts sampled, followed by *Taphozous melanopogon (58), Megaderma lyra (29), Tadarida aegyptiaca (26) Rousettus leschenaulti (22), Pipistrellus sp. (9) and Rhinopoma hardwickii (7)*. Forty-five percent of the temples had only one species, 34% had two species, 15% had three species, and 6% had four species of bats. *Hipposideros speoris* was the most common (9,715, range 2–1150) across temples followed by *R. leschenaultia* (5306, 1–2946), *T. melanopogon* (3196, 1–1165), *M. lyra* (1497, 20–605), *T. aegyptiaca* (303, 3–84), *Pipistrellus sp.* (144, 2–70) and *R. hardwickii* (50,1–15). None of the bat species we detected are "threatened" as per the International Union for Conservation of Nature (IUCN).

 **Bats and roost characteristics.** Seven species of bats in the temple used four microhabitats inside the temple (Fig 1). About 39% of the total abundance was in dark rooms, while 51% used walkways around the deity room, the main microhabitat for 5 out of 7 species. *Hipposideros speoris* occupied three out of four microhabitats, but most commonly dark rooms and walkways when dark rooms were not available or disturbed. Likewise, *M. lyra*, *T. melanopogon*

**Table 1. Species characteristics and IUCN status of bats recorded in temples.** The breeding season in parenthesis is based on the present study.

| Species name | IUCN | Diet | Breeding season | Roosting habitat |
|---|---|---|---|---|
| *Rousettus leschenaultii* | L.C. | Flowers and fruits | Nov-March | Temples, old buildings, caves |
| *Rhinopoma hardwickii* | L.C. | Moths, neuropteran insects and beetles | Feb-Mar | Crevices, roof, houses, temples, boulders |
| *Hipposideros speoris* | L.C. | Beetles, termites, low flying insects and flies | Dec-Mar (Dec) | crevices in hills, caves, caverns, disused buildings, tunnels and temples |
| *Megaderma lyra* | L.C. | Insects, reptiles, fishes and birds | Nov-Apr | caves, temples, forts, dilapidated old buildings, underground tunnels, old cow sheds, grain godowns, shallow soapstone mines and attics of houses |
| *Taphozous melanopogon* | L.C. | Insects | Jan-June (Dec-June) | ruins temples and caves, dark dungeons in the old fort |
| *Tadarida aegyptiaca* | L.C. | I Caterpillar, spider, water beetle, ground-dwelling insects | June-Sep | Crevices in cliff faces, crevices in large piles of rocks and boulders, narrow spaces formed by slabs of stone leaning against walls, the expansion joints at the top of supporting pillars in modern grain store, narrow cracks in a pillar of old mosques and crevices in buildings and old forts |
| *Pipistrellus tenuis* | L.C. | Small insects, beetle, cockroach, termite, grasshopper | Feb-march, July-Aug (Dec) | Roofs of bungalows, holes and crevices in walls, hollow branches of trees, dead leaves of trees |

and *R. hardwickii* were found in two microhabitats (walkways and crevices) while *T. aegyptiaca* was found only between pillars (crevices), and *Pipistrellus sp.* only in roof crevices. The fruit bat *R. leschenaultii* preferred walkways.

**Temple and disturbance.** The number of buildings and dark rooms significantly affected species richness and bat abundance in temples (Fig 2, Table 2). The top lmer model explained 69% of the variation in species richness, of which the fixed factors explained 58%.

Renovation of temple structures had a strong negative influence on bat species richness, while the scale of visitors to the temple negatively affected abundance (Fig 3). Species response to microhabitat and disturbance varied (Table 2). *Hipposideros speoris* abundance is influenced positively by dark rooms and the number of buildings but not by any disturbance factors. The

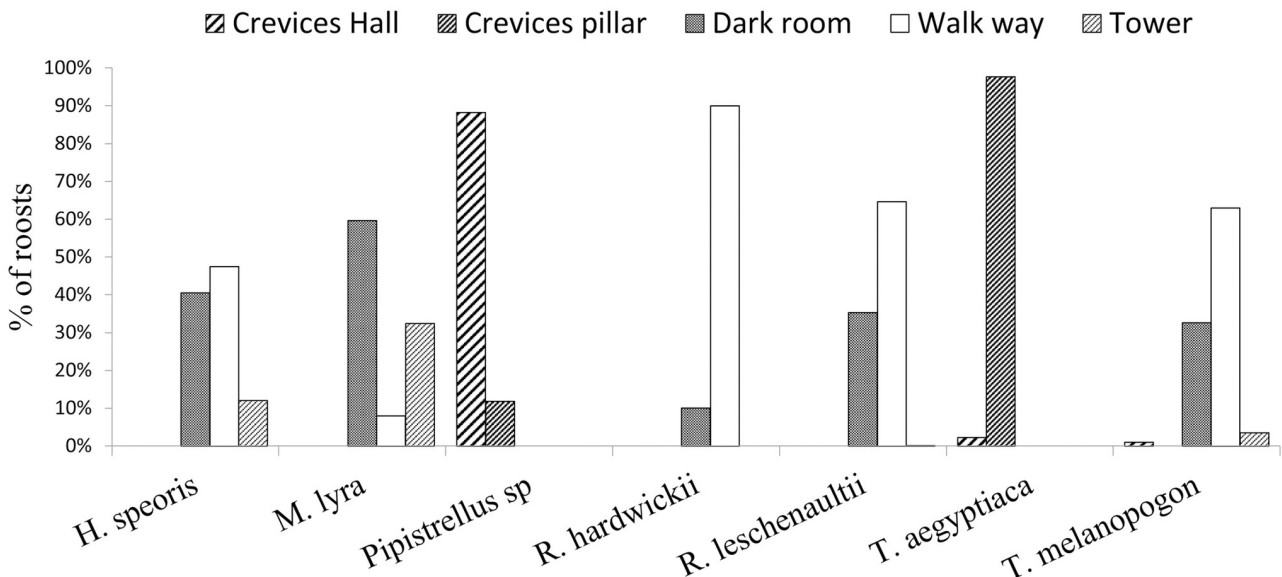

**Fig 1. Percentage of microhabitats used by the different species of bats for roosting.**

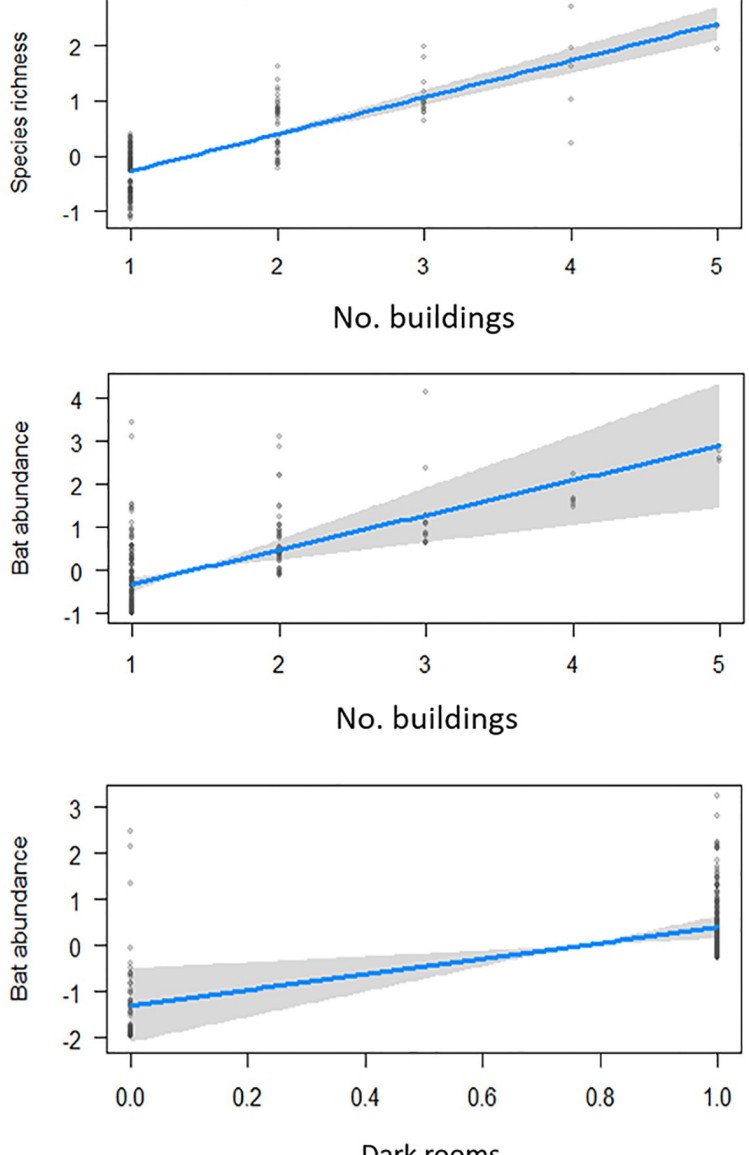

**Fig 2. Effect plots showing the relationship between species richness, abundance and the number of buildings and dark rooms in temples.**

abundance of *T. melanopogon* and *T. aegyptiaca* were also affected positively by the number of buildings but negatively by renovation. *Megaderma lyra* responded negatively to renovation, while *R. leschenaultii* was not affected by any temple characteristics or disturbance.

**Landscape.**   The percentage of land use around the temples changed with distance. At 5km from the temple, crops (63%) dominate, followed by urban areas (13%), scrub (12%), trees (7%), water (5%), grass and bare areas (0.23%). At 500m crops (54%) continued to dominate followed by urban areas (36%), scrub (0.7%), trees (3%), water (6%) grass and bare areas (0.09%). Since temples are close to settlements, urban areas dominate near the temple, and crops, trees, scrub and grass areas are fewer (Fig 4). Grasslands positively affected species richness at more than 3km from temples than other land uses, while trees in the temple complex

positively affected species richness and abundance (Table 3). As one moved away from the temple, total abundance was positively affected by scrub availability and negatively by trees, water, and urban areas. At the species level, scrub availability positively affected the abundance of *H. speoris* across the distance intervals, while trees had a negative effect (Table 3).

**Table 2. Effects of temple characteristics on species richness (lmer model) and bat abundances (glmmadmb model).** Temple id and year are included as random factors. The top models' average estimates and confidence intervals are given, and the individual model details are included as (S1 Table).

| Species richness | Estimate | Std. Error | CI 2.500 | CI 97.500 |
|---|---|---|---|---|
| (Intercept) | -0.052 | 0.122 | -0.291 | 0.187 |
| No. buildings | 0.674 | 0.044 | 0.587 | 0.761 |
| Dark rooms | 0.246 | 0.103 | 0.044 | 0.448 |
| Renovation | -0.364 | 0.081 | -0.524 | -0.204 |
| **Bat Abundances** | | | | |
| (Intercept) | 1.817 | 1.177 | -0.498 | 4.132 |
| No. buildings | 0.802 | 0.206 | 0.397 | 1.206 |
| Dark rooms | 1.597 | 0.536 | 0.541 | 2.654 |
| Walkway | 1.678 | 1.057 | -0.404 | 3.761 |
| Renovation | -0.319 | 0.354 | -1.016 | 0.378 |
| Visitors | -0.869 | 0.307 | -1.473 | -0.265 |
| Tower | 0.349 | 0.447 | -0.532 | 1.230 |
| *Hipposideros speoris* | | | | |
| (Intercept) | -0.923 | 2.296 | -5.439 | 3.594 |
| No. buildings | 0.769 | 0.345 | 0.090 | 1.448 |
| Dark rooms | 2.731 | 1.200 | 0.367 | 5.095 |
| Renovation | -0.323 | 0.602 | -1.509 | 0.863 |
| Visitors | -0.559 | 0.669 | -1.877 | 0.759 |
| Walkway | 3.631 | 2.324 | -0.948 | 8.209 |
| Tower | -0.748 | 0.922 | -2.566 | 1.069 |
| *Megaderma lyra* | | | | |
| (Intercept) | -10.230 | 3.243 | -16.610 | -3.849 |
| No. buildings | 0.982 | 0.581 | -0.162 | 2.126 |
| Renovation | -3.462 | 1.326 | -6.074 | -0.850 |
| Visitors | -1.769 | 2.205 | -6.113 | 2.575 |
| *Tadarida aegyptiaca* | | | | |
| (Intercept) | -6.086 | 3.413 | -12.775 | 0.603 |
| Renovation | -3.723 | 1.782 | -7.216 | -0.230 |
| Visitors | -2.231 | 1.942 | -6.036 | 1.575 |
| No. buildings | 1.083 | 0.479 | 0.145 | 2.021 |
| *Taphozous melanopogon* | | | | |
| (Intercept) | -14.690 | 5.538 | -25.594 | -3.786 |
| No. buildings | 0.758 | 0.318 | 0.132 | 1.384 |
| Renovation | -1.181 | 0.750 | -2.659 | 0.297 |
| Visitors | 2.312 | 1.983 | -1.594 | 6.217 |
| Dark rooms | 2.576 | 3.516 | -4.351 | 9.503 |
| *Rousettus leschenaultii* | | | | |
| (Intercept) | -9.636 | 3.318 | -16.172 | -3.100 |
| No. buildings | 1.163 | 1.248 | -1.296 | 3.622 |
| Visitors | -0.913 | 1.561 | -3.988 | 2.161 |
| Dark rooms | 0.687 | 2.822 | -4.871 | 6.245 |
| Renovation | 0.667 | 1.681 | -2.645 | 3.979 |

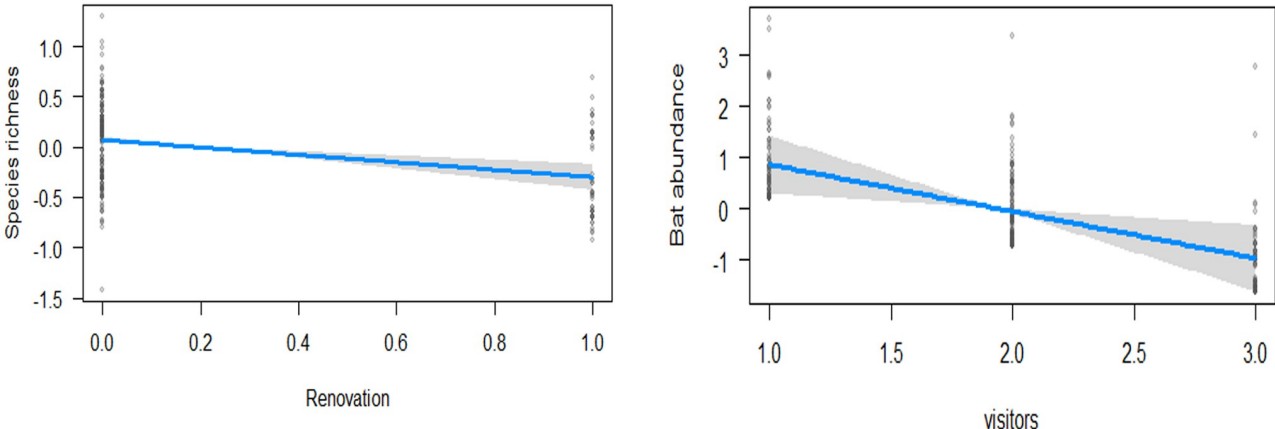

**Fig 3. Effects plots showing the relationship of disturbance (Renovation and number of visitors) on species richness and abundance.**

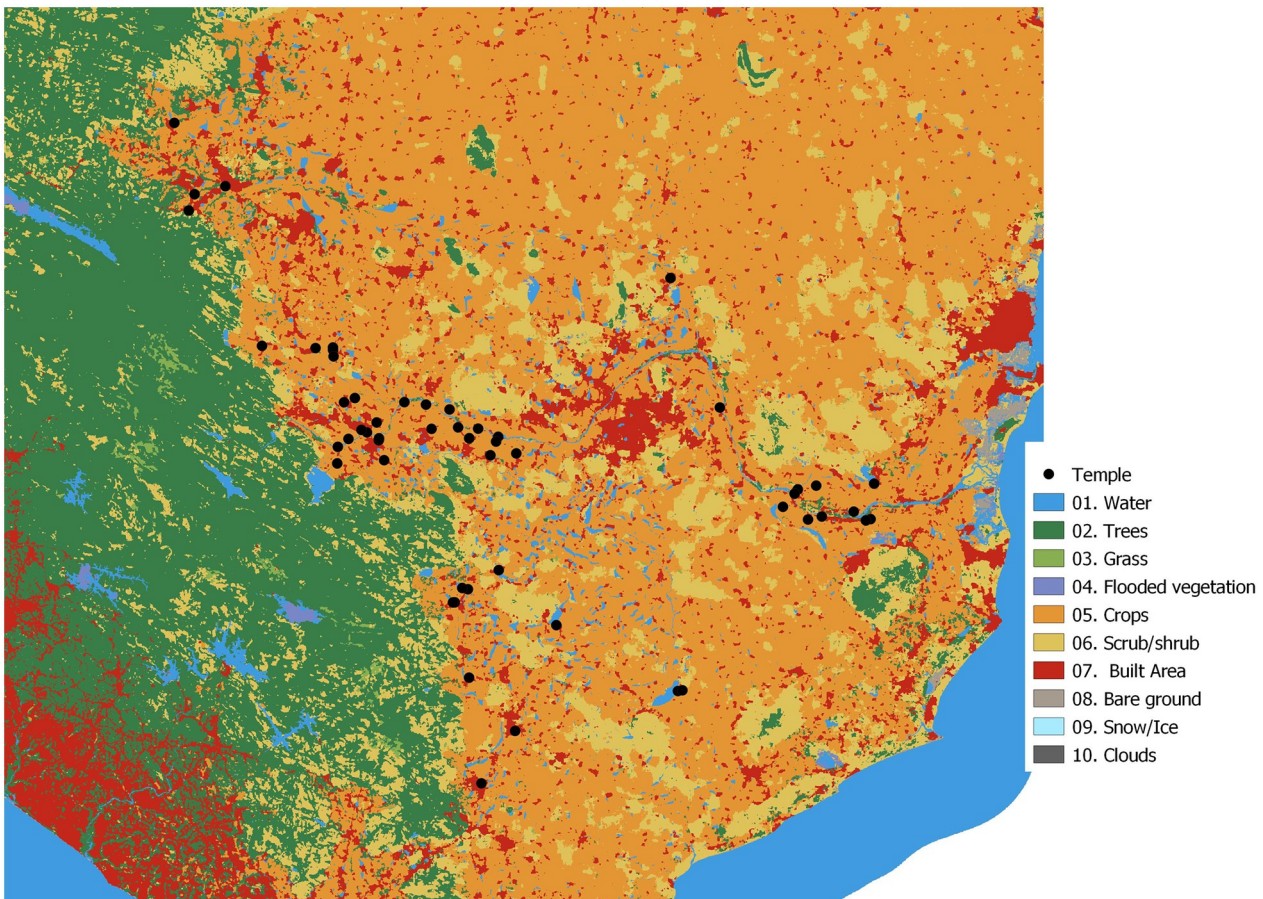

**Fig 4. High resolution (10m) land-use around the temples sampled.** Land use classification is based on Karra et al. 2021. See data analysis for more details.

**Table 3. Species response to land-use elements.** Model averaged parameter estimates of top models ($<2\,\Delta$ AICc) with non-overlapping confidence intervals are shown in the last column* The top models' details are included as (S2 Table).

| Species/Abundance | Distance (m) | Parameters | Estimate | CI 2.50% | CI 97.50% | SE | Strong association |
|---|---|---|---|---|---|---|---|
| Species richness | 0 | Intercept | 0.829 | 0.503 | 1.155 | 0.166 | |
| | | Water | 0.029 | -0.144 | 0.202 | 0.088 | |
| | | Trees | 0.006 | **0.002** | **0.010** | 0.002 | * |
| | 500 | Intercept | 0.937 | 0.694 | 1.180 | 0.123 | |
| | | Grassland | -0.071 | -0.398 | 0.255 | 0.166 | |
| | 1000 | Intercept | 0.940 | 0.695 | 1.184 | 0.124 | |
| | | Grassland | -0.197 | -0.982 | 0.588 | 0.399 | |
| | 3000 | Intercept | 0.793 | 0.541 | 1.071 | 0.128 | |
| | | Grassland | 0.653 | **0.190** | **1.120** | 0.235 | * |
| | 5000 | Intercept | 0.861 | 0.548 | 1.173 | 0.159 | |
| | | Grassland | 0.618 | -0.205 | 1.441 | 0.418 | |
| Abundance | 0 | Intercept | 2.917 | 1.963 | 3.870 | 0.484 | |
| | | Trees | 0.017 | **0.000** | **0.033** | 0.008 | * |
| | | Water | -0.174 | -0.804 | 0.456 | 0.320 | |
| | 500 | Intercept | 3.307 | 2.587 | 4.027 | 0.366 | |
| | | Trees | -0.032 | -0.097 | 0.032 | 0.033 | |
| | | Grassland | 0.184 | -1.330 | 1.698 | 0.769 | |
| | | Water | -0.001 | -0.061 | 0.059 | 0.030 | |
| | | Scrub | 0.142 | -0.088 | 0.371 | 0.116 | |
| | 1000 | Intercept | 3.229 | 1.964 | 4.495 | 0.643 | |
| | | Scrub | 0.176 | 0.057 | 0.294 | 0.060 | |
| | | Crop | 0.002 | -0.007 | 0.011 | 0.005 | |
| | | Trees | -0.144 | -0.237 | -0.051 | 0.047 | * |
| | | Urban | 0.016 | -0.018 | 0.051 | 0.018 | |
| | | Water | -0.019 | -0.089 | 0.051 | 0.035 | |
| | | Grassland | -0.094 | -3.954 | 3.767 | 1.960 | |
| | 3000 | Intercept | 3.282 | 2.007 | 4.557 | 0.648 | |
| | | Scrub | 0.066 | -0.004 | 0.136 | 0.036 | |
| | | Trees | -0.064 | -0.133 | 0.004 | 0.035 | |
| | | Grassland | 0.291 | -1.387 | 1.968 | 0.851 | |
| | | Water | -0.084 | -0.289 | 0.120 | 0.104 | |
| | | Urban | 0.035 | -0.059 | 0.129 | 0.048 | |
| | 5000 | Intercept | 7.065 | 3.306 | 10.824 | 1.913 | |
| | | Water | -0.498 | -0.864 | -0.132 | 0.186 | * |
| | | Urban | -0.139 | -0.267 | -0.010 | 0.065 | * |
| | | Trees | -0.077 | -0.137 | -0.017 | 0.030 | * |
| | | Crop | 0.001 | -0.008 | 0.011 | 0.005 | |
| | | Grassland | -0.023 | -3.133 | 3.087 | 1.579 | |
| | | Scrub | 0.069 | 0.002 | 0.137 | 0.034 | * |
| *Hiposederous speoris* | 0 | Intercept | 1.591 | 0.171 | 3.011 | 0.724 | |
| | | Trees | -0.004 | -0.032 | 0.025 | 0.015 | |
| | 500 | Intercept | 1.681 | -0.070 | 3.431 | 0.890 | |
| | | Grassland | 1.091 | -1.123 | 3.304 | 1.124 | |

*(Continued)*

**Table 3.** (Continued)

| Species/Abundance | Distance (m) | Parameters | Estimate | CI 2.50% | CI 97.50% | SE | Strong association |
|---|---|---|---|---|---|---|---|
| | | Urban | -0.036 | -0.080 | 0.008 | 0.022 | |
| | | Scrub | 0.246 | -0.128 | 0.621 | 0.190 | |
| | | Water | 0.017 | -0.085 | 0.119 | 0.052 | |
| | | Trees | -0.005 | -0.112 | 0.102 | 0.054 | |
| | 1000 | Intercept | 1.583 | -0.295 | 3.460 | 0.954 | |
| | | Scrub | 0.275 | 0.059 | 0.492 | 0.110 | * |
| | | Crop | 0.001 | -0.014 | 0.016 | 0.007 | |
| | | Trees | -0.173 | -0.333 | -0.013 | 0.081 | * |
| | | Urban | -0.026 | -0.089 | 0.037 | 0.032 | |
| | | Water | 0.015 | -0.106 | 0.135 | 0.061 | |
| | | Grassland | 0.444 | -5.734 | 6.622 | 3.136 | |
| | 3000 | Intercept | 0.672 | -0.668 | 2.012 | 0.684 | |
| | | Scrub | 0.124 | 0.009 | 0.240 | 0.059 | * |
| | 5000 | Intercept | 2.098 | -2.593 | 6.789 | 2.389 | |
| | | Grassland | -0.595 | -6.040 | 4.850 | 2.765 | |
| | | Scrub | 0.112 | -0.007 | 0.231 | 0.060 | |
| | | Water | -0.275 | -0.972 | 0.423 | 0.354 | |
| | | Urban | -0.197 | -0.446 | 0.051 | 0.126 | |
| | | Trees | -0.002 | -0.101 | 0.097 | 0.050 | |
| *Megaderma lyra* | 0 | Intercept | -10.086 | -14.399 | -6.069 | 2.219 | |
| | | Trees | 0.006 | -0.057 | 0.068 | 0.032 | |
| | | Water | -0.064 | -1.373 | 1.648 | 1.252 | |
| | 500 | Intercept | -9.165 | -12.658 | -5.673 | 1.773 | |
| | | Grassland | -1.701 | -22.185 | 18.783 | 10.399 | |
| | | Crop | -0.038 | -0.074 | -0.002 | 0.018 | |
| | | Water | -0.053 | -0.387 | 0.282 | 0.170 | |
| | 1000 | Intercept | -8.910 | -12.676 | -5.143 | 1.912 | |
| | | Grassland | -2.381 | -29.684 | 24.921 | 13.860 | |
| | | Crop | -0.031 | -0.060 | -0.001 | 0.015 | |
| | | Water | -0.130 | -0.769 | 0.509 | 0.324 | |
| | 3000 | Intercept | -9.571 | -13.736 | -5.407 | 2.115 | |
| | | Grassland | -0.325 | -9.084 | 8.435 | 4.447 | |
| | | Crop | -0.028 | -0.059 | 0.003 | 0.016 | |
| | | Water | -0.168 | -1.223 | 0.887 | 0.536 | |
| | 5000 | Intercept | -9.432 | -14.681 | -4.182 | 2.666 | |
| | | Grassland | -1.791 | -16.318 | 12.736 | 7.375 | |
| | | Crop | -0.027 | -0.061 | 0.006 | 0.017 | |
| | | Water | -0.103 | -1.593 | 1.386 | 0.756 | |
| | | Trees | 0.008 | -0.220 | 0.236 | 0.116 | |
| *Rousettus leschenaultii* | 0 | Intercept | -10.086 | -14.457 | -5.715 | 2.219 | |
| | | Trees | 0.006 | -0.057 | 0.068 | 0.032 | |
| | | Water | -0.064 | -2.531 | 2.403 | 1.252 | |
| | 500 | Intercept | -9.441 | -13.064 | -5.819 | 1.839 | |
| | | Trees | -0.354 | -1.823 | 1.115 | 0.746 | |
| | | Grassland | -2.717 | -28.261 | 22.827 | 12.968 | |

(*Continued*)

**Table 3.** (Continued)

| Species/Abundance | Distance (m) | Parameters | Estimate | CI 2.50% | CI 97.50% | SE | Strong association |
|---|---|---|---|---|---|---|---|
| | | Water | 0.029 | -0.167 | 0.224 | 0.099 | |
| | 1000 | Intercept | -9.219 | -13.146 | -5.291 | 1.994 | |
| | | Trees | -0.381 | -1.805 | 1.043 | 0.723 | |
| | | Water | 0.013 | -0.251 | 0.277 | 0.134 | |
| | 3000 | Intercept | -9.805 | -14.066 | -5.543 | 2.164 | |
| | | Trees | -0.159 | -0.928 | 0.610 | 0.391 | |
| | | Urban | 0.078 | -0.220 | 0.375 | 0.151 | |
| | | Grassland | 1.390 | -4.757 | 7.536 | 3.120 | |
| | 5000 | Intercept | -10.148 | -15.311 | -4.984 | 2.622 | |
| | | Water | 0.241 | -0.913 | 1.395 | 0.586 | |
| | | Grassland | 2.023 | -8.862 | 12.908 | 5.526 | |
| | | Trees | -0.049 | -0.391 | 0.294 | 0.174 | |
| | | Crop | -0.010 | -0.054 | 0.035 | 0.023 | |
| | | Scrub | 0.054 | -0.207 | 0.315 | 0.132 | |
| *Tadarida aegyptiaca* | 0 | Intercept | -9.608 | -14.070 | -5.146 | 2.266 | |
| | | Water | 0.555 | -1.653 | 2.763 | 1.121 | |
| | | Trees | 0.001 | -0.057 | 0.059 | 0.030 | |
| | 500 | Intercept | -8.367 | -12.152 | -4.581 | 1.922 | |
| | | Grassland | -1.819 | -19.858 | 16.221 | 9.158 | |
| | | Crop | -0.023 | -0.046 | 0.000 | 0.012 | * |
| | | Urban | -0.010 | -0.109 | 0.088 | 0.050 | |
| | | Trees | -0.317 | -1.849 | 1.215 | 0.778 | |
| | | Water | -0.085 | -0.484 | 0.314 | 0.203 | |
| | 1000 | Intercept | -7.652 | -11.239 | -4.065 | 1.821 | |
| | | Grassland | -24.772 | - 149.996 | 100.453 | 63.572 | |
| | | Crop | -0.022 | -0.043 | -0.001 | 0.010 | |
| | | Water | -0.081 | -0.506 | 0.344 | 0.216 | |
| | 3000 | Intercept | -8.235 | -13.475 | -2.995 | 2.663 | |
| | | Grassland | 0.812 | -5.570 | 7.193 | 3.240 | |
| | | Crop | -0.022 | -0.040 | -0.004 | 0.009 | |
| | | Water | -0.487 | -1.728 | 0.755 | 0.630 | |
| | | Urban | 0.061 | -0.228 | 0.350 | 0.147 | |
| | | Trees | -0.031 | -0.344 | 0.283 | 0.159 | |
| | 5000 | Intercept | -8.291 | -14.453 | -2.129 | 3.130 | |
| | | Grassland | -0.488 | -12.427 | 11.452 | 6.061 | |
| | | Crop | -0.024 | -0.044 | -0.004 | 0.010 | * |
| | | Water | -0.368 | -1.794 | 1.058 | 0.724 | |
| | | Urban | 0.117 | -0.338 | 0.572 | 0.231 | |
| | | Trees | -0.022 | -0.264 | 0.220 | 0.123 | |
| *Taphozous melanopogon* | | Intercept | -8.476 | -12.411 | -4.542 | 1.998 | |
| | | Water | -0.997 | -3.967 | 1.973 | 1.508 | |
| | | Trees | 0.010 | -0.051 | 0.071 | 0.031 | |
| | 500 | Intercept | -9.273 | -12.846 | -5.699 | 1.815 | |

(*Continued*)

**Table 3.** (Continued)

| Species/Abundance | Distance (m) | Parameters | Estimate | CI 2.50% | CI 97.50% | SE | Strong association |
|---|---|---|---|---|---|---|---|
| | | Trees | -0.096 | -0.837 | 0.645 | 0.376 | |
| | | Water | 0.029 | -0.176 | 0.233 | 0.104 | |
| | | Grassland | 0.322 | -3.117 | 3.760 | 1.746 | |
| | | Urban | 0.047 | -0.052 | 0.147 | 0.051 | |
| | | Crop | -0.007 | -0.036 | 0.021 | 0.015 | |
| | 1000 | Intercept | -9.137 | -12.374 | -5.901 | 1.644 | |
| | | Grassland | 1.637 | -5.136 | 8.410 | 3.438 | |
| | | Water | 0.056 | -0.195 | 0.308 | 0.128 | |
| | | Trees | -0.032 | -0.425 | 0.362 | 0.200 | |
| | | Crop | -0.012 | -0.030 | 0.006 | 0.009 | |
| | | Urban | 0.029 | -0.100 | 0.157 | 0.065 | |
| | 3000 | Intercept | -8.635 | -13.813 | -3.456 | 2.633 | |
| | | Grassland | 7.892 | -1.886 | 17.670 | 4.964 | |
| | | Urban | -0.208 | -0.693 | 0.276 | 0.246 | |
| | | Water | 0.302 | -0.337 | 0.940 | 0.324 | |
| | | Scrub | 0.050 | -0.162 | 0.261 | 0.107 | |
| | | Trees | -0.059 | -0.502 | 0.384 | 0.225 | |
| | 5000 | Intercept | -9.437 | -15.329 | -3.546 | 2.994 | |
| | | Grassland | 5.918 | -7.637 | 19.473 | 6.883 | |
| | | Trees | -0.091 | -0.514 | 0.332 | 0.215 | |
| | | Water | 0.198 | -0.856 | 1.252 | 0.535 | |
| | | Crop | -0.009 | -0.025 | 0.006 | 0.008 | |
| | | Urban | -0.262 | -0.930 | 0.405 | 0.339 | |
| | | Scrub | 0.081 | -0.171 | 0.333 | 0.128 | |

*Megaderma lyra* and *aegyptiaca* were negatively affected by crops up to 5km, while *T. melanopogon* and *R. leschenaultii* did not show any effect of land use on their roost abundance.

The availability of roost microhabitats across temples with and without bats was not significantly different (Kruskal Wallis test; D = 0.4, p = 0.689), Fig 5). Of the total temples sampled, 80% of them had bats across all the years, and in only 4 of 57 temples, we found bats abandoning the temple. Renovation levels were higher in temples with no bats 44% (15/34) compared with temples having bats 35% (16/46). Visitor levels were also marginally higher in no bat (x ±se, 2.11 ± 0.79) compared to temples with bats (1.89 ± 0.71). Land use between bat-occupied and not-occupied temples showed no differences except for trees in the vicinity of temples being higher in bat-occupied sites (Mann-Whitney U = 3917 p<0.01) (Fig 6), which corroborates with the land use analysis.

## Discussion

Why do bats prefer some temples? Bats occupied 80% of the temples sampled and over 20,211 individuals were counted. Though only seven species were recorded, the most common species *Hipposideros speoris* (9,715), accorded for nearly half of the population and found in more than 50% of bat-inhabited temples. *Hipposideros speoris* is a widespread species across India and Sri Lanka and occurs in large congregations with high fidelity to roost sites [48]. The other 6 species are also not rare or threatened but use many anthropogenic structures for roosting [28]. Except for *T. aegyptiaca*, *Pipistrellus sp*. and *Rhinopoma hardwickii*, species counts of all

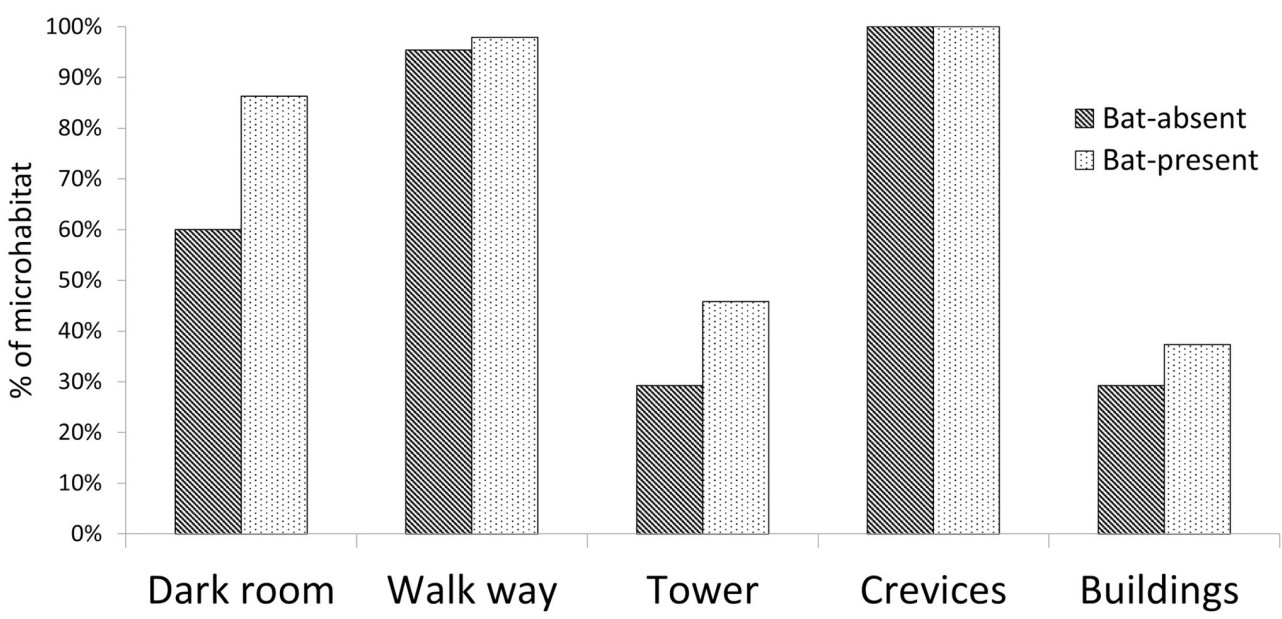

**Fig 5. Percentage availability of habitats in temples with and without bats.**

others ranged between 200 to 450 in each temple and appear to follow the general tendency of bats that roosts in caves and human-made structures to be highly gregarious [49]. Crevice roosting *T. aegyptiaca* and *Pipistrellus sp.* species may have been underestimated because they are hard to count inside the narrow crevices.

Several species that roost and breed in temples across India are cave-dwelling species that have found temples as suitable alternative habitats [28, 39]. Our results show that temples offer dark, undisturbed roosts and crevices similar to caves for seven species of insectivorous bats that account for 33% of the bat species found in the plains and hills of Tamil Nadu state [29]. Such dark areas in old temples have several openings through which bats can quickly fly out, a

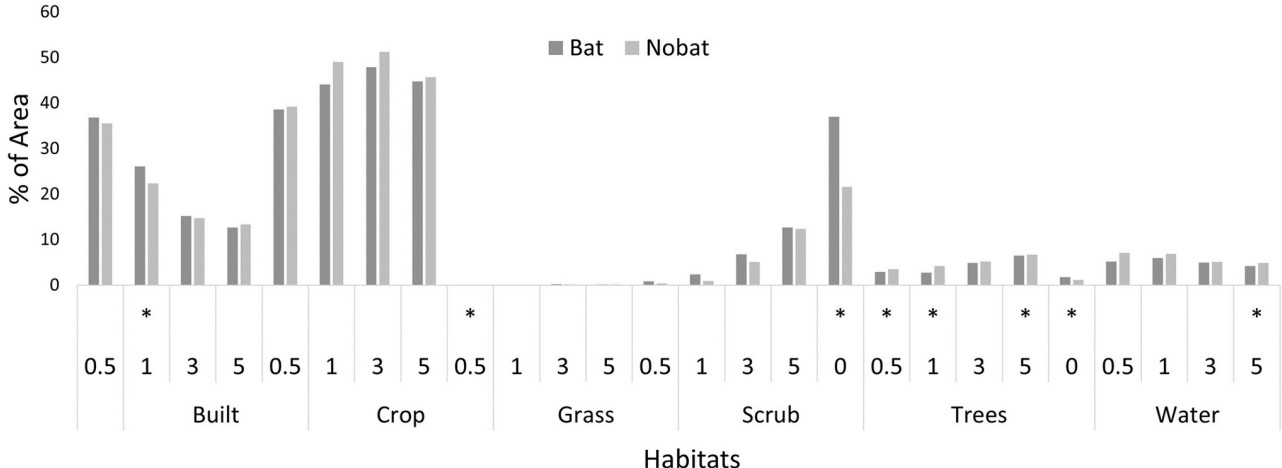

**Fig 6. Mean land-use elements around temples with and without bats at various distances.** * indicates significant differences at p< 0.05.

likely reason for their selection by most bat species. Further, granite stones used to build the ancient temples are stacked together and offer crevices for some species, such as the free-tailed bats (*T. aegyptiaca*), to roost, thereby providing a safe roost. Walkways are long corridors with a high roof with vents that remain dark during the day and provide a safe roost for bats like *H. speoris*. The high roof help bats remain unnoticed and therefore not disturbed during the morning hours when there is a movement of people. A similar response of cave-dwelling bats from the genus *Hipposideros* was found in Malaysia, where they roost in very dark corners and high places [50]. We did not take species roost height measurements, so it's not evident if species segregate roost according to the height, which is also unlikely as the height inside the temple is the same. However, unlike in a cave, many species that roost in dark corners in temples seem to segregate spatially, which varies between temples but would require more study.

This study demonstrates that disturbance to roosts is a major factor affecting roosting bats, as other similar studies have shown in old buildings under temperate conditions [20, 51]. Disturbance in the temple was less impactful on the abundant *H. speoris*, but other species were negatively affected by renovation and visitor levels. Large temples usually have more visitors but also more buildings with more potential roosts that may buffer disturbance effects. Therefore, even when disturbance levels increase, bats might persist by temporarily moving to other temple buildings. In many cases, these disturbance activities are routine in temples and bats might have become accustomed to these threats. However, several other factors that may affect bats in temples and not measured in the study, such as festivals that increase lighting, sound levels, human activity and prayer-related smoke, can impact bat abundance. The recent increase in the renovation of old temples where dark areas are lit, walls are pressure cleaned to remove the odor of bats, painted and openings 'bat proofed' to keep away bats and pigeons can effectively prevent bats from colonizing temples [48, 52]. Bats have come under even greater threat and surveillance because of local media misreporting the current coronavirus and bat association.

While temple characteristics and disturbances are significant drivers of bat abundances and occupancy, land use plays a less critical role across species. Several studies have shown the importance of landscape elements for the presence of bats [27, 53–55], but in our case, there seems to be no single habitat effect. Fine-scale (10m) mapping of land use around temples revealed differential use of habitat elements by species, but these associations were either weak or had high variations in the effect. The landscape, as mentioned earlier, is highly homogeneous; crops and urban areas together constitute about 70–80%, with scrub, grasslands, water bodies and trees comprising the rest in small proportions. Even though crops were everywhere, they did not have much effect while trees and water availability seem to have a consistent negative influence across species. The availability of scrub and grasslands is critical for species richness and the overall abundance of bats. These are the only two natural habitats available, with the rest being tree plantations or occupied by invasive *Prosopis juliflora* and artificial wetlands. Even though scrub and grasslands occupy less than 5% of the area, it has a disproportionately positive effect on all the species except *M. lyra*. The radio-tracking study in south India of the *M. lyra* showed it has a very flexible diet and can take prey both from air and ground, enabling it to use multiple habitats but prefers to forage primarily within 500m from temples [38]. The importance of natural habitat is emphasized for many species, including bats [26] and this is evident in this study. Water availability is vital for many species of bats and may drive bat movement and activity in some landscapes [25, 56]. The effect of water was variable, with species like *T. aegyptiaca* preferring water points in the temple's vicinity while others did not. Studies in Thailand on similar species have shown that insect diversity is less influential than water availability [57]. Moreover, water is not a constraint in the region because of the

perennial river and extensive irrigation network. Also, the water requirements of many of the species may be low as they are adapted to arid conditions.

Scale is vital for many species, including bats [27]. In this study, at 5km from the roost, most land-use effects became negligible, or the direction of the effects changed from positive to negative or vice versa. The extent of natural habitats such as grassland and scrub that increases with distance may influence bats to forage at larger spatial scales. For example, temples are located mainly close to human habitations and rivers while natural open dry habitats are further away and bats may be making regular forays to them, as observed in England [26].

This study provides a broad understanding of land use effect on bats. A more nuanced study and analysis will need radio telemetry and acoustic sampling to see how land-use elements influence bats. Foraging bats are most strongly associated with variables measured at small spatial scales and distance measures [27]. For instance, the only study on radio telemetry of temple bats shows 41% of the time, the bats foraged 500m from the temples but ranged up to 4km over the night during the summer months. In rainy months this may be different when the general productivity of the habitat is higher, and more extensive use of the landscape can happen as scrub and grassland habitats also become more productive. Further, with more advanced remote sensing tools, more nuanced land-use change analysis could be related to seasonal changes in bat occurrence. For example, a recent study in the same area using radar images has shown greater grassland area than conventional methods [58]; such an approach would also enable mapping across seasons irrespective of cloud cover.

The current study relies on the 2020 high-resolution land-use analysis [41], but this study predates that in 2012–2019. This mismatch, as mentioned earlier, may not be very high given the land use has remained essentially the same over this period. However, for natural habitats with less than 5% of the area and critical for bats, even small changes can significantly affect a foraging animal. Further, the dynamics of wetlands are not incorporated in this study since it was mostly done in the wet season. However, a more nuanced classification of these dynamics around each temple might be necessary as irrigation systems and water availability are affected by regional water distribution [59].

The presence of bats is linked to microhabitat or stand-level structural conditions. However, Lewis [60] found that roost-site fidelity for bats varied depending on the availability and permanency of the roosts used. For these reasons, he suggested that fidelity was high in buildings. Supporting this, a study on banded *H. speoris* bats from the study area showed 82% roost fidelity throughout the year [48]. Therefore, the temples that have remained unchanged for centuries provide conditions for high roost fidelity despite changes in the landscape and therefore are more important for bats to persist in the area.

## Conservation of bats in temples

The study has shown that microhabitats within temples, such as dark corners that are relatively less disturbed and rooms rarely used, are essential for the persistence of bats in the temples. The effect of disturbance on bats is strongly negative and can override the availability of microhabitats in temples and is also species-specific. Most of the landscape factors were surprisingly not important though the availability of natural areas around temples is crucial. While bats are known to provide critical pest control services in paddy agriculture, the role of paddy in sustaining the bat populations is not clear as crops did not have any significant effect on bat species richness and abundance. This calls for a more nuanced study using radio telemetry and acoustic sampling to understand the use of the landscape by bats since natural areas are likely to be converted to irrigated agriculture in the future. Since temples were built from breaking granite hillocks that would have harbored caves and bats in the region, temples and other old

buildings ironically are the only alternative habitats. Bat conservation in temples and historical buildings is possible if small dark corners are allowed to exist along with some abandoned areas. If this is not possible, we need to think of building alternate habitats like bat houses in the landscape and develop a comprehensive plan to let bats persist in the landscape and provide ecosystem services.

## Supporting information

**S1 Table. Top models of species abundance response to microhabitat and disturbance.** Model abbreviations-BLD: No.buildings, DR: Dark rooms, WAY: walkway, TWR: tower, REN: Renovation, VIS: visitors.
(DOCX)

**S2 Table. Top models of species response to land-use elements.**
(DOCX)

## Acknowledgments

We thank Maria, Lise, and Seshadri for help in the field. We are grateful to Dr. R. Ganesan and Dr. M.Soubadra Devy for their support during the initial stages of the work. We also thank Rajkamal, Seshadri, Prashanth and Soubadra and three anonymous reviewers for their comments on the MS.. Thamizhazhagan and Muthupandi helped us in the field while Abhishek and Thalavaipandi helped us with land use analysis. The temple authorities and staff in both Tirunelveli and Thoothukudi districts permitted us to do the study. Jermiah Rajanesan, Director, Dhonavur Fellowship and Maria Joseph, Head Master, Christuraja Hr., Sec., School, Palayamkottai, helped us with logistics, and we are grateful to them for their help. Brakes India and Sundaram Finance encouraged this work.

## Author Contributions

**Conceptualization:** T. Ganesh, M. Mathivanan.

**Data curation:** T. Ganesh, A. Saravanan, M. Mathivanan.

**Formal analysis:** T. Ganesh, A. Saravanan, M. Mathivanan.

**Funding acquisition:** M. Mathivanan.

**Investigation:** T. Ganesh, M. Mathivanan.

**Methodology:** T. Ganesh, M. Mathivanan.

**Project administration:** T. Ganesh, A. Saravanan, M. Mathivanan.

**Supervision:** T. Ganesh.

**Validation:** T. Ganesh.

**Writing – original draft:** T. Ganesh, M. Mathivanan.

**Writing – review & editing:** T. Ganesh.

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
