## [Decision Letter · Decision Letter 0]

16 Jun 2021

PONE-D-21-13889

Temples the last abode for bats in a homogeneous agriculture matrix: Importance of microhabitat availability and land use factors for bat conservation.

PLOS ONE

Dear Dr. T,

Thank you and your co-authors for submitting your manuscript to PLoS ONE. In this regard, we solicited 3 reviewers for adjudication, and all three agreed. The reviews were remarkably consistent (despite some disagreement in the ultimate recommendation) in their agreement that the manuscript is germane to the volume, and will build on existing knowledge of bats utilizing built structures. All three reviewers were positive on the topic and scope, and the authors should be commended.

However, the reviewers also were largely in agreement that considerable work remains to render this manuscript suitable for publication in PLoS ONE. Reviewer 1 and 3 provided a thorough reviews arguing for "Major Revisions," while Reviewer 2, also providing an in-depth review, argued for minor revisions. Despite this difference in opinion, the reviewers were in agreement on two problem areas of the manuscript, one cosmetic and one substantive. From a cosmetic standpoint, all reviewers (and having read the manuscript themselves, the AE concurs) agree that the manuscript requires substantial editing to correct considerable errors in grammar and syntax. It is important to recognize that these errors do not detract from the quality of the research, and are beyond the scope of PLOS ONE's criteria for publication. Nevertheless, the degree and volume are extensive, and thus revision is non-trivial.

The substantive issue, however, is more serious. The reviewers agree and the AE concurs that there is a degree of opacity in the analytical approaches that require elucidation before the manuscript can be fully vetted (at least with respect to the analyses, their results, and the inferences drawn from them). The reviewers identify a number of areas in the analytical methods that require additional detail and clarification. Given PLoS ONE's publication criteria (specifically that "(e)xperiments, statistics, and other analyses are performed to a high technical standard and are described in sufficient detail," and "(c)onclusions are presented in an appropriate fashion and are supported by the data"), it is critical that the methods are much more clearly defined and defended. Doing so will allow reviewers to better assess whether or not the analyses were appropriate, and the veracity of the results they deliver.

Given the above, the Academic Editor, in a conservative approach, summarized the opinions of the reviewers as a petition for "Major Revisions."

We look forward to receiving your revised manuscript.

Kind regards,

Mark A. Davis, Ph.D.

Academic Editor

PLOS ONE

Journal Requirements:

3. In your Methods section, please provide additional location information, including geographic coordinates for the data set if available.

"NO"

"No"

6. We note that you have indicated that data from this study are available upon request. PLOS only allows data to be available upon request if there are legal or ethical restrictions on sharing data publicly. For more information on unacceptable data access restrictions, please see http://journals.plos.org/plosone/s/data-availability#loc-unacceptable-data-access-restrictions.

7. We note that Figure 1 in your submission contain map images which may be copyrighted. All PLOS content is published under the Creative Commons Attribution License (CC BY 4.0), which means that the manuscript, images, and Supporting Information files will be freely available online, and any third party is permitted to access, download, copy, distribute, and use these materials in any way, even commercially, with proper attribution. For these reasons, we cannot publish previously copyrighted maps or satellite images created using proprietary data, such as Google software (Google Maps, Street View, and Earth). For more information, see our copyright guidelines: http://journals.plos.org/plosone/s/licenses-and-copyright.

7.1.    You may seek permission from the original copyright holder of Figure 1 to publish the content specifically under the CC BY 4.0 license. 

7.2.    If you are unable to obtain permission from the original copyright holder to publish these figures under the CC BY 4.0 license or if the copyright holder’s requirements are incompatible with the CC BY 4.0 license, please either i) remove the figure or ii) supply a replacement figure that complies with the CC BY 4.0 license. Please check copyright information on all replacement figures and update the figure caption with source information. If applicable, please specify in the figure caption text when a figure is similar but not identical to the original image and is therefore for illustrative purposes only.

8. Please include your tables as part of your main manuscript and remove the individual files. Please note that supplementary tables should be uploaded as separate "supporting information" files.

Reviewers' comments:

Reviewer's Responses to Questions

**Comments to the Author**

1. Is the manuscript technically sound, and do the data support the conclusions?

Reviewer #1: Partly

Reviewer #2: Yes

Reviewer #3: Partly

2. Has the statistical analysis been performed appropriately and rigorously? 

Reviewer #1: No

Reviewer #2: Yes

Reviewer #3: Yes

3. Have the authors made all data underlying the findings in their manuscript fully available?

Reviewer #1: Yes

Reviewer #2: Yes

Reviewer #3: Yes

4. Is the manuscript presented in an intelligible fashion and written in standard English?

Reviewer #1: No

Reviewer #2: Yes

Reviewer #3: Yes

5. Review Comments to the Author

Reviewer #1: There is some clarification needed to determine if statistics were statistically sound. Including which data were used in which analysis. The authors sampled 59 temples repeatedly, and it is unclear if they treated each survey as an independent sample, or if data from only one survey year were used, or if it was averaged across the 5 survey periods for each temple.

There are multiple grammatical and spelling errors throughout that do not detract from the quality of research, but would need to be corrected before publication.

Overall this study would be a valuable contribution to the literature on bats using anthropogenic structures, and my specific comments for improving the manuscript are attached.

Reviewer #2: Overall, this paper has significant importance for publishing by emphasizing the importance of temples as habitat worthy of conservation especially at a time when bats are at a high risk of exclusion from places because of the pandemic. Some modifications are necessary to clarify the paper and its findings, however they do not detract from the importance of the study.

Reviewer #3: Specific suggestions in attached document.

Methodology needs explanation - 59 temples surveyed over 5 years would be 295 surveys. Not sure how the seasonality is included. Were temples surveyed over the years in each season? How did year influence species richness and abundance? Were morning and afternoon counts added together?

Needs more grammatical review - some misspellings, incomplete and run-on sentences

Passive voice is used throughout manuscript

Tables need consistent font, decimal places and headers. Explanation of 2.5 and 97.5 needed in table 3.

Results could use more explanation of output values from analysis.

Figures need major edits - proper labeling of variables, Figure 6 has no y-axis, land use x-axis should be 0, 0.25, 0.50, 0.75, 1

Why on Figure 5 and 6 are species abundance and richness only 10-30? Did you only use means? Using raw data would be much more informative.

6. PLOS authors have the option to publish the peer review history of their article (what does this mean?). If published, this will include your full peer review and any attached files.

Reviewer #1: No

Reviewer #2: No

Reviewer #3: No

---

## [Author Response · Author response to Decision Letter 0]

7 Dec 2021

All responses to reviewers are marked as authors response(AR) in red.

Editor

The reviewers agree and the AE concurs that there is a degree of opacity in the analytical approaches that require elucidation before the manuscript can be fully vetted (at least with respect to the analyses, their results, and the inferences drawn from them). The reviewers identify a number of areas in the analytical methods that require additional detail and clarification.

AR: We have re written the Material and methods section giving clarity in the methods used and the data used in the analysis 

the manuscript requires substantial editing to correct considerable errors in grammar and syntax.

AR: This has been addressed to the extent possible

Reviewer 1.

Review: Temples the last abode for bats in a homogeneous agriculture matrix: Importance of

microhabitat availability and land use factors for bat conservation.

Ganesh et al studies historic temples as roosting habitat for bats in India. This work highlights the importance of anthropogenic structures as essential roosts, particularly as temples are places where humans are actively working to remove bats. While focusing on one region in India, the authors conducted roost counts and identified species of bats over several years while also assessing each temple for several microhabitat and landscape factors. They conducted linear models to compare species richness and abundance to the landscape and microhabitat variables and calculated niche breadth and overlap. The authors found seven species of bat across forty seven out of fifty-four temples that they surveyed. Dark rooms and corridors were found to be the most important microhabitats for roosting, while landscape effects had little overall effect albeit with some species-specific instances. Bat abundance varied by species in season, but overall was lowest during the monsoon season. There was little species overlap in temples and little niche overlap. The authors attributed a lot of the microhabitat preferences to species-specific life history and roosting preferences. The seasonality differences correlated with the parturition and lactation of many of the species. While the landscape effect had a weak response, the importance of scrub and open landscape is still noted. Overall, this paper lays the foundation for further studies and emphasizes the importance of an anthropogenic roost for bats. This paper certainly merits publication with additional clarification and editing. 

My first concern overall for the paper is some grammatical and formatting issues. I have listed specific examples of reoccurring problems throughout at the end of this review, however there are additional grammar edits that need to be fixed. 

My second concern is with the landscape ecology analysis and discussion throughout the paper. The analysis of the landscape features was measuring presence/absence of landscape features over a significantly sized area. This broad measuring does not get at some of the heterogeneity of the landscape and may explain some of the lack of effect of the landscape features in the model. A more rigorous measurement of the landscape types could be better such as using the total amount or percentage of land cover type in the 5km buffer. As the remote sensing data may be unable to provide this level of detail (as mentioned in the methods), a mention of this drawback in the discussion would be warranted. Additionally, the discussion section touched on the landscape feature briefly but the middle of the paragraph still discussed temple characteristics such as festivals and renovations and the ending sentences focused on site fidelity. The importance of open spaces was stated but never explained or elaborated on. A deeper discussion with a paragraph focused solely on landscape characteristics while moving the still-relevant topics to more appropriate places would improve the analysis especially since the third hypothesis on landscape features was disproven.

Overall, this paper has significant importance for publishing by emphasizing the importance of temples as habitat worthy of conservation especially at a time when bats are at a high risk of exclusion from places because of the pandemic. Some modifications are necessary to clarify the paper and its findings, however they do not detract from the importance of the study.

Authors response(AR): 

The landscape analysis has been completely revised. We have used recently released ESA landcover classes at 10m resolution for the analysis (https://tiledimageservices.arcgis.com/P3ePLMYs2RVChkJx/arcgis/rest/services/Esri_2020_Land_Cover_V2/ImageServer). However, this approach allows us to estimate the landcover class for a particular period and not across seasons because of cloud cover as mentioned in the MS earlier. We have therefore assumed, based on government literature that landcover has not drastically changed during the study period which is also substaintiated by authors observations. We analysis the % landcover under each class to species richness and abundance of bats at 3 spatial scales;500m, 1km and 3km. The variable in the landcover class is crop which we determine based on google earth images as earlier and interviews with farmers around temples. Some periods of the sampling May and Sep do not have any crops and there fore we have considered crop area in such times as zero and considered it as fallow agriculture. More details in the MS. 

Reviewers: A deeper discussion with a paragraph focused solely on landscape characteristics while moving the still-relevant topics to more appropriate places would improve the analysis especially since the third hypothesis on landscape features was disproven.

AR: Our discussion has changed and improved given the changes in the analysis. We have as suggested given more space for discussing landscape characterstics.

Grammatical and Formatting Comments:

Check the spacing throughout the article. There should be a space in between the last word and opening parentheses. On the last sentence of page 11, there should be a space after the comma and between each year. The same issues occur in the caption for Table 4 with the list of symbols. Lastly, there should be a space in between genus and species in the species names in the results - I.e. M. lyra, not M.lyra. 

In the first two paragraphs of the introduction, there are a few words and several citations where the year is in blue. They should all be in black since they are not hyperlinks. 

A w is bolded in the word “with” in the beginning of the full paragraph on page 10. 

Throughout the entire paper, halls is followed by (corridors) or vice versa each time. I think saying “halls and corridors” is appropriate the first time and then use one consistently for the rest of the paper. 

AR: We agree on the English and have tried our best to improve. Some of the issues have also arised due to formatting. 

We have modified halls to a more appropriate word ‘walkway’

Figure 1 appears blurry, so I would recommend the resolution to be larger. Partially because of the resolution, the temples and town dots are hard to tell apart. It would be easier to read if one of the dots was hollow. 

AR: We have redone the figure

Figure 3 does not have axis labels which helps the reader.

Corrected

In Figure 4, the dark room and hall patterns are very hard to tell apart between the bars. A different pattern would make the graph easier to read. 

We have redone the figure

Figure 5 has extra ticks around the outside of the graph. 

We have redone the figure

In Figure 8, the no-village bar is distracting and confusing as it overlaps with the error bars. 

Changed the fig as we are using landcover maps

Reviewer 2

Reviewer Summary of Paper: This is a study of factors important for bats using ancient temples and overlap of roost use in bats using ancient temples. The authors investigate roost characteristics of temples (microhabitat) and surrounding land use (landscape-scale habitat at 5 km scale). They determine that roost characteristics (mainly presence of dark rooms and halls) influence bat richness/abundance in temples. Landscape variables at the 5 km scale were not important for bat abundance and bat presence. 

Overall impression: The subject of managing historic/ancient buildings and bat conservation will appeal to many readers. The study would add to the literature of managing ancient/historic buildings for bats. However, the manuscript would need major revision before publication in PLoS One. The discussion is the strongest section of this paper, and other sections could be improved by incorporating more literature into the introduction to set up the hypotheses, clarifying the hypotheses, organizing the data analysis section by hypothesis, adding more information about which data were used in which analysis, and clarifying some of the figures. 

My specific comments to the authors to improve the manuscript are outlined below:

Title: 

1. The term “matrix” implies a heterogenous landscape of multiple land use/cover types not a homogenous one (only one cover type). I suggest rewording or clarifying the title. 

Author response AR: replaced “matrix” with “landscape”

Abstract:

2. Please specify which pandemic.

3. AR: Mentioned as COVID19 pandemic

4. “We sampled 59 temples repeatedly across 5 years which yielded a sample of 256 temple.” I believe that 59 temples sampled repeatedly is a sample size of 59 temples. I suggest saying “which yielded 256 survey events” or rephrasing another way. Was every temple sampled 5 times?

AR:Yes sample size is 59 temples and these were repeatedly sampled. Suitably modified in the abstract. The number of times each temple was sampled varied between 1 to 5. Have mentioned the methods. 

Introduction: 

The introduction should be supported with more literature to justify the research question. I suggest the authors incorporate the literature on bats using anthropogenic structures and introduce and explain the issues here for bats, humans managing structures, and bat-human conflicts. A good paper to read is Fagan et al. 2018. “Roost selection by bats in buildings, Great Smoky Mountains National Park” in the Journal of Wildlife Management 82: 424-434, as they evaluated bat roost selection of historic buildings in a non-urbanized area and also measured landscape variables. If the literature is limited for bats in buildings, the authors could draw on the literature of bats using bridges or other anthropogenic transportation structures that humans have to manage. One example is Meierhofer et al. 2019. “Structural and environmental predictors of presence and abundance of tri-colored bats in Texas culverts” in the Journal of Mammalogy 100: 1274-1281. One of the more interesting aspects of this subject is that bats using historical buildings or other human infrastructure makes it difficult to balance bat conservation with human-bat conflicts. And this could be applied beyond ancient temples. 

More information on the types of bat species that use these temples, why we should care about them, and current conservation status would greatly improve this manuscript. For example, the authors could mention the importance of bats to agricultural pest suppression (if these bats are insectivores) or seed dispersal (if these bats are frugivores) or providing other examples of ecosystem services. The authors should explain what they mean by the pandemic negatively affecting bats. Future readers may not know which pandemic the authors are referring to, and why it affected bats. If any species are “data deficient” or of conservation concern on the IUCN Red List of Threatened Species (https://www.iucnredlist.org/), this should also be mentioned in this introduction. Additionally, if there have been previous studies involving these species, i.e., what they usually use to roost if not temples, what they eat, where they forage, etc. this would be helpful to include some basic background information to set up the hypotheses. Are they mostly fruit bats or are they insectivores? Some species background information is provided in the discussion but is really needed in the introduction to support the justification of these research questions. 

Similarly, more information on temple structure, e.g., What materials are they made out of? How do bats get into the temples (through doors or cracks in the walls or unknown?) Are they air conditioned or are they about the same temperature/hotter/cooler than outside?) Additionally, introduce readers to the monsoon seasons briefly in the introduction and more in the study area section. 

AR: More information on temple structure given in the MS itself

The authors should be careful about using the term “niche” when only referring to roosting habitat. A niche encompasses all needs of a species (roosting habitat, foraging habitat, all biotic and abiotic factors (climate), etc.). In some parts of this manuscript, I believe “niche” should be replaced with “roost habitat”. 

The objectives of this paper need to be clarified if they are investigating what microhabitat and landscape factors determine roost use and bat abundance in order to inform bat conservation in ancient temples, and/or if they are investigating which bats have roost overlap needs (what I think the authors are referring to when they say “niche overlap and breadth”). What is important about quantifying niche overlap/breadth?

AR: we have removed niche analysis from the results.

5. “These are situated in a human-dominated agricultural matrix that is changing due to several economic reasons and climate change effects (Fischer et al. 2009, 2010; Jones et al. 2009).” Please describe the climate change effects affecting bats in this area. Do you mean the monsoons? I also suggest including literature on anthropogenic disturbance, agriculture, and habitat/fragmentation threats to support your justification for this research. I suggest developing those ideas more with Frick et al. 2019 “A review of the major threats and challenges to global bat conservation” paper (available at: https://www.researchgate.net/publication/332146982_A_review_of_the_major_threats_and_challenges_to_global_bat_conservation ) or other studies 

AR: The role of climate and habitat transformation in the region has been stated in the context of Frick et al 2019 paper. We have also added some available literature for the region.

6. “The recent pandemic has further made them more vulnerable to disturbance and direct persecution.” Please describe which pandemic and why this is an issue for bats. 

AR: COVID 19, Done

7. “ 1. Given that the temple bats are mostly cave-dwelling species we expect larger

Yinpterochiroptera occupying similar micro-habitat, show greater niche overlap and similar niche breadth.” Please introduce the species before stating the hypotheses. This is the first time the bats and some traits are mentioned. Please describe niche breadth and overlap in this context. Does this mean the authors are trying to determine overlap in roost needs/preferences of species using the temples and that they expect Yinpterochiroptera to have overlapping needs with each other? Why does this matter? Please clarify this hypothesis. Are Yinopterochiroptera the only group that use the temples? Are they all cave-dwelling? 

AR: we have modified this hypothesis. We have removed niche overlap from the paper. Mention of species is done earlier and most of the species are insectivorous expect for one 

8. “2. Microclimate conditions inside the temples are very stable and any influence on

bat abundances could be a reflection of food availability or disturbance at roost and therefore we expect bats to show seasonal changes in niche overlap and breadth due to variability in crop and insect availability.” 

This hypothesis is confusing to me. I’m not sure what you are hypothesizing. Would it not be better to test abundance changes seasonally rather than niche overlap and breadth? The climate/monsoon seasons could also be described before introducing this hypothesis. Additionally, it is unclear how disturbance at roost and food availability were measured. 

AR: we have modified this hypothesis and have made a separate result for roost disturbance which is mentioned in the introduction and methods.

9. “3. Since the structure of ancient temples are all similar, we do not expect any

change in microhabitat availability in temples with and without bats, instead we expect bat abundances driven by the availability of landscape features especially seasonal crops around the temples.” Please describe how the temples differ or are similar to each other and the seasonality of crops before introducing this hypothesis. Also, is it not true that the temples are structurally different? Some are smaller/larger and more complex than others? 

AR:We have described the temples in detail and the type of crop and their availability in the section. Temple did vary in size but structurally they were more or less similar. 

Materials and methods:

“The ancient temples(>400 years) are built of large granite stones and provide a niche for several bat species.” I think the authors mean that the temple structures provide roost habitat, not a full niche. 

AR: Agreed and we have clarified in the text 

“The surveyed temples are situated in an agricultural matrix surrounded by rice paddies or bananas” Are the temples completely surrounding in the whole 5 km radius landscape or just mostly surrounded in the immediate area around the temple? What is the extent of the surrounding agriculture? An agricultural matrix implies mixed cover types so this sentence is somewhat confusing.

AR: The entire area even beyond 5km are dominated by paddy. We have given more information about the landscape in the introduction and study area section

“stone pillars that supported the stone roof was considered a separate habitat as they provided a niche for few species that roost in the gaps created between the stone joints this was categorized as crevices” This sentence is a little confusing as it reads. Were the stone joints in the pillars categorized as crevices? The stone pillars provided a roost, not a niche. 

AR: Yes, they provide crevices and therefore a roost habitat

“In 2012 we measured temperature and humidity while sampling for bats and found no difference between temples and therefore not considered in the analysis.” What are the temperature and humidity conditions in these temples and is it similar to cave conditions used by these bats?

AR: We are not aware of temperature and humidity conditions inside caves in the region as caves are non-existent because the rocky hills have been quarried. Conditions inside the temples are mentioned though, as stated in the methods section, we could not monitor these continuously 

“Since we did not have information on how far the different bat species forage in the landscape we took 5 km based on the study on the large bat Megaderma lyra (Audet et al 1991).” Did the authors consider testing multiple spatial scales? Many bat studies have determined that landscape scale factors are important at different scales. See the discussion in Bellamy et al. 2013 “Multiscale, presence-only habitat suitability models: fine resolution maps for eight bat species” in the Journal of Applied Ecology 50: 892-901 (available at: https://besjournals.onlinelibrary.wiley.com/doi/full/10.1111/1365-2664.12117) about importance of factors at multiple landscape scales. In the case of not knowing foraging distance for all species, it would improve this study to quantify landscape availability at more than one scale to determine if it is important (For example, maybe test 500 m, 1 km, and 5 km or landscape scales that are relevant to bats in other parts of the world). What kind of bat is the Megaderma lyra? Is it an insectivorous species? 

AR: Thank you for the paper link. We have used one-time remote sensing classification of the land use available at ESA website and reanalyzed the data at 4 different spatial scales as suggested. Since such information was available only at one particular year for the entire region we considered the land use not to have changed during the sampling period and when the image was taken. This we justify based on government records of area under agriculture. We also tried to do a fine level classification by gridding the area around temples and finding the land use within the grid cells using google earth. This was a herculean task as the grid cells for each temple at 3km was over 1000 and was not possible to do that for each temple across years and is the reason we resorted to remote sensing analysis for fine scale analysis

.

AR: Megaderma lyra is an insectivorous species and we have mentioned the details of all the species in a separate Table 1..

 “If a land use element is present we gave a value of 0.25 in each quadrant and calculated the frequency of each land use for the entire 5km radius area.” Is 25% (i.e, 0.25) sensitive enough to detect variation? Why 25% and not 10%? If a 5 km radius landscape only has a little bit of banana, but it is present, then it would still get a value comprising 25% of the landscape? This may be a limitation of the study to mention in your discussion. Would the results be different if the 5 km landscape was divided by 10% increments? 

AR: This is a very valid point and we agree on the bias and as mentioned above we have done our best under the current conditions. Keeping all the comments related to landuse we have changed the analysis as mentioned before. 

Did land use around each temple differ or was it correlated among the 5 sampling years? 

AR: Yes, the land use did not change much across years

“The best model was selected using.” Please finish this sentence and indicate what AICc weights and delta AIC value criteria were used to determine plausible models. 

AR: done

“We constructed a linear model to explain the species richness and abundance of bats in the temples based on the microhabitat and landscape features. We based the variables used in our linear models on our observations made in 2012.” Does this mean that only the landscape features from 2012 and the bat counts from 2012 were used? I think it would be statistically incorrect if all survey years at the same temples were treated as independent temples in the models. 

AR:We have use mixed model approach to repeated sampling of temples across years following Bolker et al. 2009 wherein they say “Repeated measurements on the same individual, the same location, or observations taken at the same point in time are often correlated; this correlation can be accounted for by using random effects in generalized linear mixed models”. Since there were variations both within the sampled temples and across years, making temple and years as random factors accounts for this variation and gives a more robust relationship between the response variable (species richness and abundance of bats) with explanatory variables like temple features and land use elements. 

Please organize the data analysis section to be more clear which analysis you are doing to answer which question/hypothesis and clarify which data were used to examine microhabitat and landscape factor influence on abundance and richness. In other words, did you use only data from 2012 or did you treat each repeated survey as a separate “temple”? If only used 2012 data, what were the monsoon conditions at that time? 

AR: Temples were sampled for its features in 2012 and if any new temples were added in later years their features was recorded then. Temple features are subjected to modifications and therefore we have included renovation of temples and visitors to the temple as variables that could affect bats. Renovation is a dynamic variable that can differ between years and therefore we have measured it in all sampled years in all temples compared to more static ones like number of buildings, tower etc which do not change across years. 

We have now tried to bring in more clarity in the data analysis section as suggested 

Results:

“The total number of roosts sampled was 351 across 246 temples in 5 years.” This sentence is a little confusing. There were only 59 temples. Were any of the roosts that were repeatedly sampled the same roosts as previous years? 

AR: We have modified the sentence and yes, same roosts were sampled across years and some temples also had multiple roosts

“None of these bat species is threatened as per IUCN category.” Are any of them considered “data deficient”? Please move this species information to the introduction to introduce the species that use the temples.

AR: Done. No bat was data deficient 

“The lmer model explained 67% of the variation in species richness of which 55% were explained by the fixed factors.” Do you mean the model averaged mixed model comprising all models with AICc <5 or the top model or something else? 

AR: The % values pertain to the top model. The models have changed after reanalysis. The delta AICc was kept at <2 unless mentioned specifically. We have corrected for any discrepancies 

Discussion:

The authors are effective at relating the results to the previous literature and this is the strongest section of the paper.

Mention the limitations of the study (e.g., temples were chosen based on logistics rather than randomly, only one landscape scale was considered, land use in the surrounding 5 km radius was considered at a course resolution (0.25)) 

AR: The landscape section has been reanalyzed completely as mentioned earlier

From your results, are their certain species that may be more impacted by temple renovation/disturbance? 

AR: We have now made renovation as an objective as mentioned earlier and identify species that are affected, most species are.

Figures:

Fig 1 map is a bit blurry. I suggest increasing dpi/resolution. Did there appear to be any spatial pattern of temples that did not have bats vs. those that did? If so, mention in discussion.

AR: The fig has been redone. We did not test for spatial correlations 

Fig 4 bar graph gray colors are difficult to differentiate between dark room and hall.

AR: Corrected

Fig 5 is this a bar graph of all temples, including the ones that were resampled? If so, perhaps plot the average species richness across the 5 years against the no. of buildings in each temple. It is difficult to interpret as is. 

AR: These are not bar graphs, they are effect plots from the mixed model showing the relationship between number of buildings and species richness keeping other factors constant in the model. Such effect plots are routinely used to plot the results of mixed models and are considered valid to predict the relationships between variables.

We have corrected the graph and included datapoints along with the prediction line. 

Same comment as last figure (which is also labeled Fig 5. Is that intentional?) What is DR? Define in figure or in parentheses in caption.

AR: expanded and clarified

Fig 6 I suggest the axis title say “Proportion of rice paddy in 5 km radius” or similar

AR: After the new analysis using remote sensing categorization we could not do the analysis at individual crop level so this is excluded 

Fig 8 This figure is difficult to read with No. villages plotted on top of the land use classes. 

AR: Changed completely

Table 1: Nicely presented. Please indicate in the caption at what values are there considered an overlap? Higher values mean more overlap?

AR: We have removed this analysis 

Table 4: Missing a variable after the “+” under Hiposederous speoris? Did you mention that you also had species-specific models in the main text? 

AR: Modified

Reviewer 3

Made comments in the MS itself which is addressed therein

---

## [Decision Letter · Decision Letter 1]

10 Mar 2022

PONE-D-21-13889R1Manuscript Click here to access/download;Manuscript;Bats in ancient temples of South India.docxPLOS ONE

Dear Dr. T,

Thank you for submitting your manuscript to PLOS ONE. After careful consideration, we feel that it has merit but does not fully meet PLOS ONE’s publication criteria as it currently stands. Therefore, we invite you to submit a revised version of the manuscript that addresses the points raised during the review process.

All three original reviewers agreed to review this re-submission. All three reviewers agree and I concur that the substantive issues have been adequately addressed to yield a vastly improved manuscript. Nevertheless, a number of grammatical, typological, and formatting issues remain. This manuscript requires a very careful proofing and editing process that addresses these issues. Commendably, the reviewers have provided extensive edits to improve readability. Given that there are no substantive issues with the manuscript, if the authors are able to address the readability of the manuscript, upon receipt of a revised version that has corrected the editorial issues, I will move to accept for publication without any further external review.

We look forward to receiving your revised manuscript.

Kind regards,

Mark A. Davis, Ph.D.

Academic Editor

PLOS ONE

Journal Requirements:

Additional Editor Comments (if provided):

Thank you for your revisions and a vastly improved manuscript. All reviewers agree that the substantive issues have been well-addressed and that the manuscript is nearly ready for publication. However a number of grammatical, typological, and formatting errors persist that must be addressed prior to publication.

Reviewers' comments:

Reviewer's Responses to Questions

**Comments to the Author**

1. If the authors have adequately addressed your comments raised in a previous round of review and you feel that this manuscript is now acceptable for publication, you may indicate that here to bypass the “Comments to the Author” section, enter your conflict of interest statement in the “Confidential to Editor” section, and submit your "Accept" recommendation.

Reviewer #1: All comments have been addressed

Reviewer #2: (No Response)

Reviewer #3: (No Response)

2. Is the manuscript technically sound, and do the data support the conclusions?

Reviewer #1: Yes

Reviewer #2: Yes

Reviewer #3: (No Response)

3. Has the statistical analysis been performed appropriately and rigorously? 

Reviewer #1: Yes

Reviewer #2: Yes

Reviewer #3: (No Response)

4. Have the authors made all data underlying the findings in their manuscript fully available?

Reviewer #1: No

Reviewer #2: Yes

Reviewer #3: Yes

5. Is the manuscript presented in an intelligible fashion and written in standard English?

Reviewer #1: No

Reviewer #2: Yes

Reviewer #3: (No Response)

6. Review Comments to the Author

Reviewer #1: It is obvious that the authors have spent a lot of time and effort in revising this manuscript. The details on the analyses have been provided and the methods are sound. The information presented is very interesting. There is not a data availability statement in the manuscript, but the authors do indicate in the submission form that they are available upon request. There are still many grammatical errors, but I have provided specific suggestions for the authors to improve the grammar and readability. My suggestions are included in the attached PDF and the authors can use Adobe Acrobat Reader DC to view the edits.

Reviewer #2: Review: Temples the last abode for bats in a homogeneous agriculture landscape: Importance of microhabitat availability, disturbance and land use factors for bat conservation.

The edits to this article made the methods much more robust and clarified many of my previous concerns, especially the updated methods for landcover classification. There remain some spacing and grammatical issues some of which I lined out below. With minor grammatical and editorial revisions, this paper will meet publication criteria.

Revisions

Ensure there is a space between every genus and species- M. lyra, not M.lyra

Ensure there is a space between each paratheses and last word in each instance- i.e. temples (Audet et al 1991) or trees (3%)

Figure 6’s font is quite hard to read. I think the resolution of the image needs to be increased.

Double check the style of your citations. The format differs between citations with respect to spacing and comma use between the last name and year.

In the last paragraph, two species mentioned need to have the Genus capitalized and the species italicized.

The last sentence in the Bats and Roost Characteristics paragraph is incomplete or contains an extra word.

The list of years- 2012,2013,2014,2018,2019. – should be 2012, 2013, 2014, 2018, 2019 with spaces in between and no period at the end. Similar spacing should be maintained in the list of distances surveyed in the Data Analysis paragraph.

I would recommend rephrasing your title and your introductory sentence of the Conservation of bats in temples paragraph. You’ve shown that many bats consistently utilize temples as roosts and that they have preferred microhabitats, but I don’t think you can infer that temples are essential for the persistence of bats. To do this, you would need telemetry data/population-level data for species to show that temples are the majority of roosting locations and reducing temples reduces bat populations because of limited roosting which may not be true for common species. There may be enough other buildings nearby to sustain bat populations without temples.

Reviewer #3: There are still many grammatical corrections to make for publication (attached Word document with track changes). Also the in-text citations are not consistent with regard to semicolons and commas between citations and periods after et al. There are still species abbreviations missing the space between genus and species as mentioned by reviewer 1 previously.

7. PLOS authors have the option to publish the peer review history of their article (what does this mean?). If published, this will include your full peer review and any attached files.

Reviewer #1: No

Reviewer #2: No

Reviewer #3: No

---

## [Author Response · Author response to Decision Letter 1]

13 Jun 2022

Review Comments to the Author

Reviewer #1: It is obvious that the authors have spent a lot of time and effort in revising this manuscript. The details on the analyses have been provided and the methods are sound. The information presented is very interesting. There is not a data availability statement in the manuscript, but the authors do indicate in the submission form that they are available upon request. There are still many grammatical errors, but I have provided specific suggestions for the authors to improve the grammar and readability. My suggestions are included in the attached PDF and the authors can use Adobe Acrobat Reader DC to view the edits.

Authors response (AR)

Thank you for the reviews and edits on the manuscript. We have incorporated your edits and have gone through the MS thoroughly to correct for grammatical errors.

Reviewer #2: Review: Temples the last abode for bats in a homogeneous agriculture landscape: Importance of microhabitat availability, disturbance and land use factors for bat conservation.

The edits to this article made the methods much more robust and clarified many of my previous concerns, especially the updated methods for landcover classification. There remain some spacing and grammatical issues some of which I lined out below. With minor grammatical and editorial revisions, this paper will meet publication criteria.

Revisions

Ensure there is a space between every genus and species- M. lyra, not M.lyra

Ensure there is a space between each paratheses and last word in each instance- i.e. temples (Audet et al 1991) or trees (3%)

AR:We have corrected for space

Figure 6’s font is quite hard to read. I think the resolution of the image needs to be increased.

AR:We have changed the resolution. The actual image is clearer than the embedded one. 

Double check the style of your citations. The format differs between citations with respect to spacing and comma use between the last name and year.

In the last paragraph, two species mentioned need to have the Genus capitalized and the species italicized.

AR:We have used the Plos One reference format

The last sentence in the Bats and Roost Characteristics paragraph is incomplete or contains an extra word.

The list of years- 2012,2013,2014,2018,2019. – should be 2012, 2013, 2014, 2018, 2019 with spaces in between and no period at the end. Similar spacing should be maintained in the list of distances surveyed in the Data Analysis paragraph.

AR: Rectified

I would recommend rephrasing your title and your introductory sentence of the Conservation of bats in temples paragraph. You’ve shown that many bats consistently utilize temples as roosts and that they have preferred microhabitats, but I don’t think you can infer that temples are essential for the persistence of bats. To do this, you would need telemetry data/population-level data for species to show that temples are the majority of roosting locations and reducing temples reduces bat populations because of limited roosting which may not be true for common species. There may be enough other buildings nearby to sustain bat populations without temples.

AR: We agree. Accordingly have modified the title “Temples and bats in a homogeneous agriculture landscape: Importance of microhabitat availability, disturbance and land use for bat conservation”. 

Reviewer #3: There are still many grammatical corrections to make for publication (attached Word document with track changes). Also the in-text citations are not consistent with regard to semicolons and commas between citations and periods after et al. There are still species abbreviations missing the space between genus and species as mentioned by reviewer 1 previously.

AR: We have now addressed these issues. We have removed species abbreviations.

---

## [Editor Report · Decision Letter 2]

22 Jun 2022

Temples and bats in a homogeneous agriculture landscape: Importance of microhabitat availability, disturbance and land use for bat conservation

PONE-D-21-13889R2

Dear Dr. T,

We’re pleased to inform you that your manuscript has been judged scientifically suitable for publication and will be formally accepted for publication once it meets all outstanding technical requirements.

Kind regards,

Mark A. Davis, Ph.D.

Academic Editor

PLOS ONE

Additional Editor Comments (optional):

Thank you for your diligence in revising this manuscript. While a few minor errors in grammar, syntax, and formatting remain, I cam confident that they can be dealt with in the proofs. I look forward to seeing this article published.
---

## [Editor Report · Acceptance letter]

30 Jun 2022

PONE-D-21-13889R2 

Temples and bats in a homogeneous agriculture landscape: Importance of microhabitat availability, disturbance and land use for bat conservation 

Dear Dr. T:

I'm pleased to inform you that your manuscript has been deemed suitable for publication in PLOS ONE. Congratulations! Your manuscript is now with our production department. 

Kind regards, 

on behalf of

Dr. Mark A. Davis 

Academic Editor

PLOS ONE